# Efficacy of anti-vascular endothelial growth factor agents for treating neovascular age-related macular degeneration in vitrectomized eyes

**Yongseok Mun** [1,2], **Kyu Hyung Park**[1,2], **Sang Jun Park** [1,2], **Se Joon Woo** [1,2] *

**1** Department of Ophthalmology, Seoul National University College of Medicine, Seoul, South Korea, **2** Seoul National University Bundang Hospital, Seongnam, South Korea

* sejoon1@snu.ac.kr

## Abstract

### Purpose

To evaluate the efficacy of intravitreal anti-vascular endothelial growth factor (anti-VEGF) agents for treatment of neovascular age-related macular degeneration (nAMD) in vitrectomized eyes.

### Methods

The medical records were reviewed of nAMD patients treated with anti-VEGF agents who previously underwent pars plana vitrectomy (PPV). PPV was performed with complete posterior vitreous detachment induction.

### Results

A total of 44 eyes from 44 patients were included. The mean central foveal thickness (CFT) was 478.50 ± 156.93 μm at baseline, 414.25 ± 143.55 μm (86.6% of baseline) at 1 month after first injection (P < 0.001), and 386.75 ± 141.45 μm (80.8% of baseline) after monthly multiple injections (2.30 ± 1.07; range, 1–5) (P < 0.001). The mean logarithm of the minimum angle of resolution best-corrected visual acuity visual acuity (BCVA) was 0.85 ± 0.57 at baseline, 0.86 ± 0.63 after the first injection, and 0.84 ± 0.64 after monthly multiple injections. BCVA improved in 39.5% at 1 month after first injection and 45.2% at 1 month after monthly multiple injections. In the subgroup analysis, CFT of eyes with the posterior capsule decreased significantly to 85.8% and 79.8% of baseline values at 1 month after the first injection and after monthly multiple injections, respectively. CFT of eyes without the posterior capsule decreased to 91.6% and 87.4% of baseline values at 1 month after the first injection and after monthly multiple injections, respectively, without statistical significance.

**Data Availability Statement:** All relevant data are within the paper.

**Funding:** This study was supported by the National Research Foundation (NRF) grant 2020R1F1A1072795, the NRF Bio & Medical Technology Development Program (Grant No. 2018M3A9B5021319) funded by the Korean government (MSIT) and a research grant from Seoul National University Bundang Hospital (13-2019-003). The funders had no role in study design, data collection and analysis, decision to publish, or preparation of the manuscript.

**Competing interests:** The authors have declared that no competing interests exist.

## Conclusion

Monthly injections of Intravitreal anti-VEGF agents induced favorable anatomical improvement and vision maintenance in vitrectomized eyes with nAMD.

## Introduction

Age-related macular degeneration (AMD) is a leading cause of visual impairment and blindness worldwide [1]. The advent of intravitreal anti-vascular endothelial growth factor (VEGF) injection led to a paradigm shift of treatment for neovascular AMD (nAMD) [2,3]. Bevacizumab (Avastin; Genentech, Inc., South San Francisco, CA, USA), ranibizumab (Lucentis; Genentech, Inc.), and aflibercept (Eylea; Regeneron Pharmaceuticals, Tarrytown, NY, USA) are anti-VEGF agents used for the treatment of nAMD, for which favorable results in non-vitrectomized eyes were reported through clinical trials [4–6].

In real-world clinical practice, the retinal diseases that require vitrectomy such as retinal detachment, vitreous hemorrhage, complicated cataract surgery, macular hole, or macular epiretinal membrane, are usually more prevalent in the elderly than in the young people [7–12]. Patients who undergo vitrectomy may also need intravitreal anti-VEGF injections for the treatment of nAMD.

However, it is uncertain whether intravitreal anti-VEGF injection effectively treats nAMD in vitrectomized eyes since all clinical trials excluded cases of pars plana vitrectomy (PPV). Most retinal specialists argue that the intraocular clearance of intravitreal anti-VEGF would increase and therapeutic efficacies would be substantially reduced after vitrectomy, but firm evidence of this is lacking. Previous studies reported controversial findings on the pharmacokinetics of intravitreal anti-VEGF agents after vitrectomy in animal eyes [13–17]. Several clinical studies reported on the effect of intravitreal anti-VEGF in vitrectomized eyes, but these were only small case series (nAMD, 4 eyes), cohorts of patients with pathologies other than nAMD (macular edema due to diabetic retinopathy or retinal vein occlusion), or cases of patients who underwent core rather than total vitrectomy [18–23].

This study aimed to evaluate the efficacy of intravitreal anti-VEGF agents for the treatment of nAMD in vitrectomized eyes in a larger number of patients to reach a clinically significant conclusion.

## Materials and methods

We retrospectively reviewed the electronic medical records of 141 vitrectomized eyes of 141 patients with nAMD treated between March 2009 to December 2018. No patient had bilateral eyes with nAMD after vitrectomy. We obtained approval from the Institutional Review Board (IRB) of Seoul National University Bundang Hospital (no. B-1905/543-111) and conducted the study in accordance with the tenets of the Declaration of Helsinki. The IRB approved a waiver of informed consent.

The inclusion criteria were: (1) history of vitrectomy; (2) history of intravitreal anti-VEGF injection for treatment of nAMD after vitrectomy; and (3) a minimum follow-up of 1 month after intravitreal anti-VEGF injections. The exclusion criteria were: (1) other causes of choroidal neovascularization (CNV), for example, pathologic myopia, uveitis, trauma, angioid streak; (2) combination treatment consisting of photodynamic therapy (PDT) and anti-VEGF injection in the same day; (3) no anti-VEGF injections despite macular fluid—i.e., proactive

treatment; and (4) concurrent other retinal diseases that caused macular edema such as diabetic retinopathy and retinal vein occlusion. Since eyes with breakthrough vitreous hemorrhage (VH) could not be evaluated, non-treatment-naïve eyes with breakthrough VH were enrolled at the time of the first anti-VEGF injection after resolution VH.

Before the first anti-VEGF injection after vitrectomy, each patient underwent ophthalmologic examinations, including slit lamp biomicroscopy, measurement of best-corrected visual acuity (BCVA) with a Snellen chart and intraocular pressure, indirect ophthalmoscopy, spectral-domain optical coherence tomography (Spectralis OCT; Heidelberg Engineering, Heidelberg, Germany), and fluorescein and indocyanine green angiography (HRA-2; Heidelberg Engineering). Two cases without BCVAs were excluded from the visual outcome analysis.

PPV was performed using a Constellation vision system or Accurus surgical system (Alcon Laboratories, Inc., Forth Worth, TX, USA) which were used for both 23- or 25-gauge. Complete posterior vitreous detachment (PVD) and peripheral vitrectomy were performed following core vitrectomy in all cases. The vitreous cortex was removed as much as possible.

Anti-VEGF agents including bevacizumab, ranibizumab, and aflibercept were administered aseptically and prophylactic topical antibiotics were applied after the injection [24]. The First anti-VEGF injections were administered to treatment-naïve eyes within 2 weeks after the diagnosis of nAMD. The first Anti-VEGF injections were administered to non-treatment-naïve eyes within 2 weeks after macula fluid was observed. The treatment protocol was identical for vitrectomized and non-vitrectomized patients: 3 monthly loading injections were administered initially to treatment-naïve patients. Thereafter, patients received anti-VEGF injections pro re nata or according to a treat-and-extend regimen at the physician's discretion. In the treatment-naïve group, eyes with breakthrough VH did not receive anti-VEGF at the end of vitrectomy. The treatment response was monitored by optical coherence tomography (OCT). Based on OCT findings, patients who showed an insufficient response were administered repeated injections monthly until a sufficient decrease in intraretinal or subretinal fluid, i.e., fluid-free macula, was achieved.

The main outcome was central foveal thickness (CFT) measured by OCT at 1 month after the first anti-VEGF injection and 1 month after monthly multiple injections. BCVA was collected with the CFT and converted to the logarithm of the minimum angle of resolution (logMAR) before the analysis. Counting fingers and hand motions were converted to 1.98 and 2.28 in logMAR, respectively [25]. The total number of anti-VEGF injections in 1 year and BCVA 1 year after the first injection were also collected. Improvement and worsening of BCVA were defined as a change $\geq 0.1$ LogMAR from baseline. Subgroup analyses were performed by comparing patients with and without posterior capsule as well as treatment-naïve patients and non-treatment-naïve patients.

The statistical analysis was performed using SPSS software (version 20.0.0; SPSS, Inc., Chicago, IL, USA). A paired t-test and the Wilcoxon signed-rank test were used to compare CFT and logMAR before versus after treatment. Analysis of variance (ANOVA) was used to compare CFT difference according to CNV type. All assumptions for ANOVA use were met, as the three groups were normally distributed ($P > 0.05$, Shapiro-Wilk test) and the variances among them were even ($P > 0.05$, Levene test). Values of $P < 0.05$ were considered significant.

## Results

A total of 44 eyes of 44 patients were included in the analysis. The patient flow is displayed in Fig 1. The patients' demographics are described in Table 1. The mean patient age at vitrectomy was $71.16 \pm 7.41$ years; 24 patients (54.5%) were men. Table 1 shows the baseline characteristics of the included patients and eyes. Of the 44 eyes, 21 (47.7%) required vitrectomy for

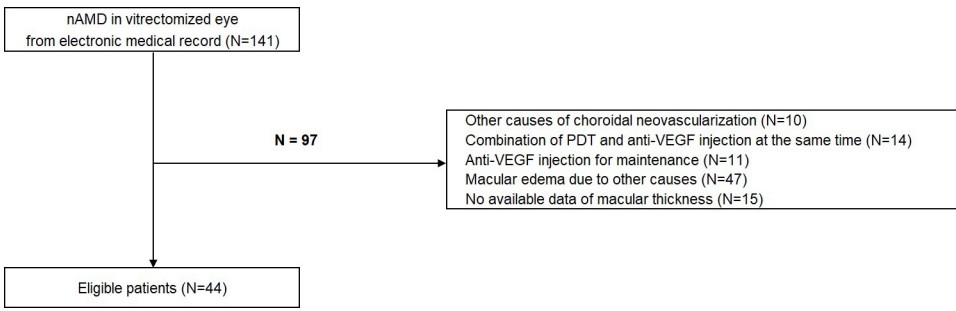

**Fig 1. Flow chart of patient selection process.** nAMD = neovascular age-related macular degeneration;
VEGF = vascular endothelial growth factor; PDT = photodynamic therapy.

breakthrough VH and subretinal hemorrhage secondary to nAMD. Vitrectomy was also needed
in 13 eyes (29.5%) for epiretinal membrane, 5 (11.4%) for posterior capsular rupture, 2 (4.5%)
for macular hole, 1 (2.3%) for rhegmatogenous retinal detachment, 1 (2.3%) for intraocular lens
dislocation, and 1 (2.3%) for vitreomacular traction. Thirty-four eyes (77.3%) had type 1 CNV.
Type 2 and 3 CNV were seen in 6 eyes (13.6%) and 3 eyes (6.8%), respectively. CNV type was
uncertain in 1 eye (2.3%). Thirty-seven eyes (84.1%) had pseudophakia, 29 eyes (65.9%) were
given intravitreal anti-VEGF injection within 3 months after vitrectomy and 25 eyes (56.8%)
were given less than three intravitreal anti-VEGF injections serially. Twenty-seven eyes (61.4%)
were non-treatment-naïve and 2 eyes (4.5%) received PDT before enrolling in our study. The
total mean anti-VEGF injections was 4.14 ± 2.12 in 1 year after the first injection.

Fig 2 shows representative images of OCT and fundus fluorescein angiography. This patient
was diagnosed with nAMD at 3.6 years after vitrectomy for rhegmatogenous retinal detach-
ment and had a successful initial response to anti-VEGF; the CFT was 237 μm (53.7% of base-
line). After three injections, the CFT decreased to 224 μm (50.8% of baseline).

The Mean CFT was 478.50 ± 156.93 μm at baseline, 414.25 ± 143.55 μm (86.6% of baseline)
at 1 month after the first injection, and 386.75 ± 141.45 μm (80.8% of baseline) at 1 month
after multiple monthly (2.30 ± 1.07; range, 1–5) injections. Mean CFT was decreased signifi-
cantly at 1 month after the first injection and monthly multiple injections (P < 0.001) (Table 2
and Figs 3 and 4). Changes in CFT were not significantly associated with CNV or anti-VEGF
agent type at 1 month after the first injection or monthly multiple injections (Tables 3 and 4).

In patients with BCVA information (n = 42), mean LogMAR BCVA was 0.85 ± 0.57 at base-
line, 0.86 ± 0.63 at 1 month after the first injection, and 0.84 ± 0.64 at 1 month after monthly
multiple injections. There was also no significant difference in BCVA between pre- and post-
injection (P = 0.985 and 0.911, respectively) (Fig 5). Mean LogMAR BCVA was 0.83 ± 0.61 at 1
year after the first injection. Of the 38 eyes (those of 4 eyes were missing), BCVA was improved
in 15 eyes (39.5%), maintained in 12 eyes (31.6%), and deteriorated in 11 eyes (28.9%) at 1
month after the first injection. Of the 42 eyes, BCVA was improved in 19 eyes (45.2%), main-
tained in 14 eyes (33.3%), and deteriorated in 9 eyes (21.4%) at 1 month after monthly multiple
injections.

In the subgroup analysis of the posterior capsule, the mean CFT of eyes with posterior cap-
sule (n = 38) decreased significantly after the first injection (P < 0.001) and monthly multiple
injections (P < 0.001). There was no significant change in BCVA (n = 36) after the first injec-
tion (P = 0.714) or monthly multiple injections (P = 0.942). The mean CFT of eyes without
posterior capsule (n = 6) did not significantly improve at 1 month after the first injection
(P = 0.917) or monthly multiple injections (P = 0.249). There was no significant difference in
BCVA after the first injection (P = 0.141) or monthly multiple injections (P = 0.785) (Table 5).

**Table 1. Demographic and baseline characteristics of the 44 study participants.**

| | | Total (N = 44) | With posterior capsule (N = 38) | Without posterior capsule (N = 6) | Treatment-naïve (N = 17) | Non-treatment-naïve (N = 27) |
|---|---|---|---|---|---|---|
| Age: vitrectomy (years) | | 71.16 ± 7.41 | 71.87 ± 7.34 | 66.67 ± 6.77 | 69.82 ± 6.85 | 72.00 ± 7.75 |
| Age: first anti-VEGF injection (years) | | 72.39 ± 6.75 | 72.68 ± 7.08 | 70.50 ± 4.04 | 72.47 ± 4.95 | 72.33 ± 7.76 |
| Sex (male:female) | | 24:20 | 21:17 | 3:3 | 8:9 | 16:11 |
| Diabetes mellitus | | 29.5% (13) | 28.9% (11) | 33.3% (2) | 35.3% (6) | 25.9% (7) |
| Hypertension | | 43.2% (19) | 44.7% (17) | 33.3% (2) | 35.3% (6) | 48.1% (13) |
| Other systemic disease | Hyperlipidemia | 9.1% (4) | 10.5% (4) | 0.0% (0) | 11.8% (2) | 7.4% (2) |
| | Cancer | 9.1% (4) | 7.9% (3) | 16.7% (1) | 17.6% (3) | 3.7% (1) |
| Cause of vitrectomy | RRD | 2.3% (1) | 2.6% (1) | 0.0% (0) | 5.9% (1) | 0.0% (0) |
| | Intraocular lens dislocation | 2.3% (1) | 0.0% (0) | 16.7% (1) | 5.9% (1) | 0.0% (0) |
| | Macular epiretinal membrane | 29.5% (13) | 34.2% (13) | 0.0% (0) | 35.3% (6) | 25.9% (7) |
| | Breakthrough VH | 34.1% (15) | 39.5% (15) | 0.0% (0) | 29.4% (5) | 37.0% (10) |
| | Subretinal hemorrhage | 13.6% (6) | 15.8% (6) | 0.0% (0) | 11.8% (2) | 14.8% (4) |
| | Vitreomacular traction syndrome | 2.3% (1) | 2.6% (1) | 0.0% (0) | 0.0% (0) | 3.7% (1) |
| | Posterior capsular rupture | 11.4% (5) | 0.0% (0) | 83.3% (5) | 5.9% (1) | 14.8% (4) |
| | Macular hole | 4.5% (2) | 5.3% (2) | 0.0% (0) | 5.9% (1) | 3.7% (1) |
| Choroidal neovascularization type | Type 1 | 77.3% (34) | 78.9% (30) | 66.7% (4) | 82.4% (14) | 74.1% (20) |
| | Type 2 | 13.6% (6) | 10.5% (4) | 33.3% (2) | 5.9% (1) | 18.5% (5) |
| | Type 3 | 6.8% (3) | 7.9% (3) | 0.0% (0) | 5.9% (1) | 7.4% (2) |
| | Unknown | 2.3% (1) | 2.6% (1) | 0.0% (0) | 5.9% (1) | 0.0% (0) |
| Lens status | Phakia | 15.9% (7) | 18.4% (7) | 0.0% (0) | 5.9% (1) | 22.2% (6) |
| | Pseudophakia | 84.1% (37) | 81.6% (31) | 100.0% (6) | 94.1% (16) | 77.8% (21) |
| Interval between vitrectomy and first anti-VEGF injection | <3 mo | 65.9% (29) | 71.1% (27) | 33.3% (2) | 47.1% (8) | 77.8% (21) |
| | 3–6 mo | 4,5% (2) | 5.3% (2) | 0.0% (0) | 0.0% (0) | 7.4% (2) |
| | 6–12 mo | 2.3% (1) | 2.6% (1) | 0.0% (0) | 0.0% (0) | 3.7% (1) |
| | >1 year | 27.3% (12) | 21.1% (8) | 66.7% (4) | 52.9% (9) | 11.1% (3) |
| Number of monthly multiple injections | 1 | 29.5% (13) | 31.6% (12) | 16.7% (1) | 11.8% (2) | 40.7% (11) |
| | 2 | 25.0% (11) | 26.3% (10) | 16.7% (1) | 17.6% (3) | 29.6% (8) |
| | 3 | 34.1% (15) | 28.9% (11) | 66.7% (4) | 64.7% (11) | 14.8% (4) |
| | 4 | 9.09% (4) | 10.5% (4) | 0.0% (0) | 0.0% (0) | 14.8% (4) |
| | 5 | 2.27% (1) | 2.63% (1) | 0.0% (0) | 5.9% (1) | 0.0% (0) |
| First anti-VEGF agent | Bevacizumab | 43.2% (19) | 47.4% (18) | 16.7% (1) | 29.4% (5) | 51.9% (14) |
| | Ranibizumab[a] | 29.5% (13) | 28.9% (11) | 33.3% (2) | 35.3% (6) | 25.9% (7) |
| | Aflibercept | 27.3% (12) | 23.7% (9) | 50.0% (3) | 35.3% (6) | 22.2% (6) |

VEGF = vascular endothelial growth factor; RRD = Rhegmatogenous retinal detachment; VH = vitreous hemorrhage.

[a]One patient received 3 bevacizumab injections after 1 ranibizumab injection and the other patient received 2 bevacizumab injections after 3 ranibizumab injections due to insurance coverage.

In the subgroup analysis of treatment history, a significant decrease in CFT was observed in treatment-naïve eyes (n = 17) after the first injection (P<0.001) and multiple monthly injections (P < 0.001). However, there was no significant change in BCVA (n = 16) after the first injection (P = 0.181) or monthly multiple injections (P = 0.071). In non-treatment-naïve eyes (n = 27), a significant decrease in CFT was also observed after the first injection (P = 0.002) and monthly

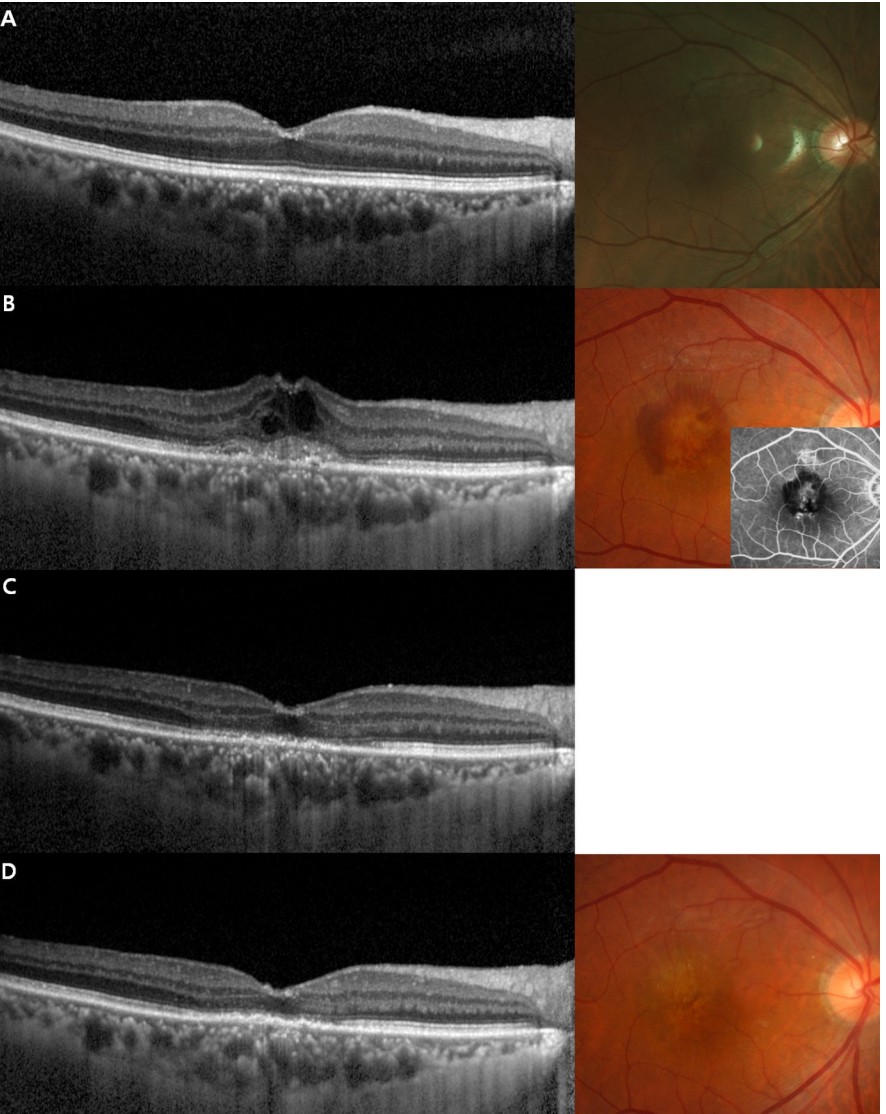

**Fig 2. Representative images of a patient who responded to intravitreal anti-vascular endothelial growth factor (VEGF) agents after vitrectomy.** (A) Optical coherence tomography (OCT) at 2 years after vitrectomy without neovascular age-related macular degeneration. (B) OCT and fluorescein angiography showing an active choroidal neovascular membrane at 3.6 years after vitrectomy. (C) Mild intraretinal fluid was observed at 1 month after the first anti-VEGF injection. (D) OCT showed a dry retina (absence of intraretinal or subretinal fluid) at 1 month after three serial injections.

**Table 2. Comparison of central foveal thickness (CFT) after anti-VEGF injection.**

|  | Baseline | 1 month after first injection | 1 month after monthly multiple injection |
|---|---|---|---|
| CFT (mean ± SD) | 478.50 ± 156.93 μm | 414.25 ± 143.55 μm (86.6% of baseline value) | 386.75 ± 141.45 μm (80.8% of baseline value) |
| Change in thickness | N/A | 64.25 ± 78.00 μm | 91.75 ± 85.68 μm |
| P value | N/A | <0.001[a] | <0.001[a] |

[a]Paired t-test.

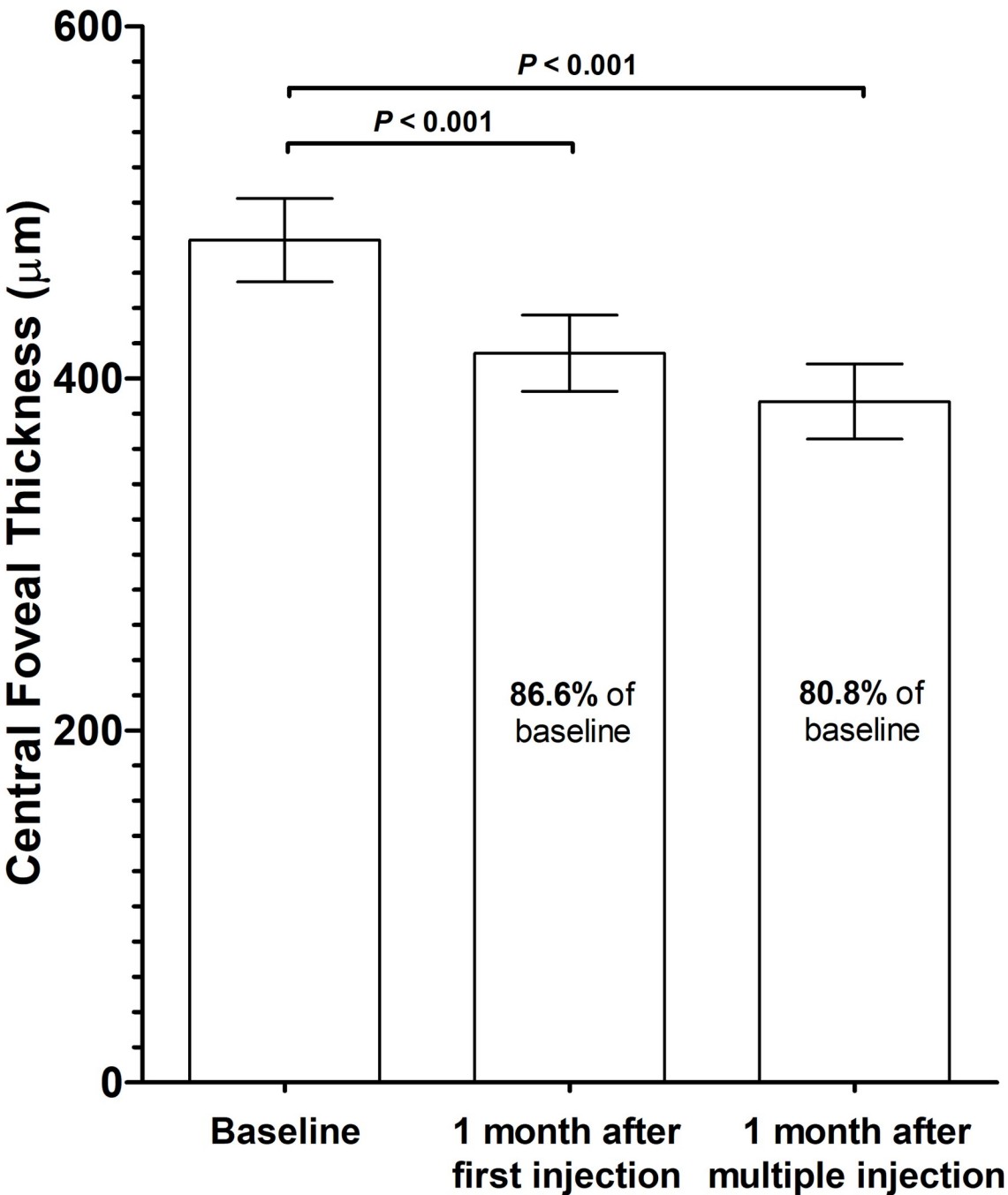

**Fig 3. Change in central foveal thickness (CFT).** The bars represent mean CFT and the error bars represent standard error of mean.

multiple injections (P < 0.001); however, there was no significant change in BCVA (n = 26) after the first injection (P = 0.384) or monthly multiple injections (P = 0.423) (Table 5).

## Discussion

Our study showed that intravitreal anti-VEGF injections in vitrectomized nAMD eyes decreased CFT and maintained BCVA.

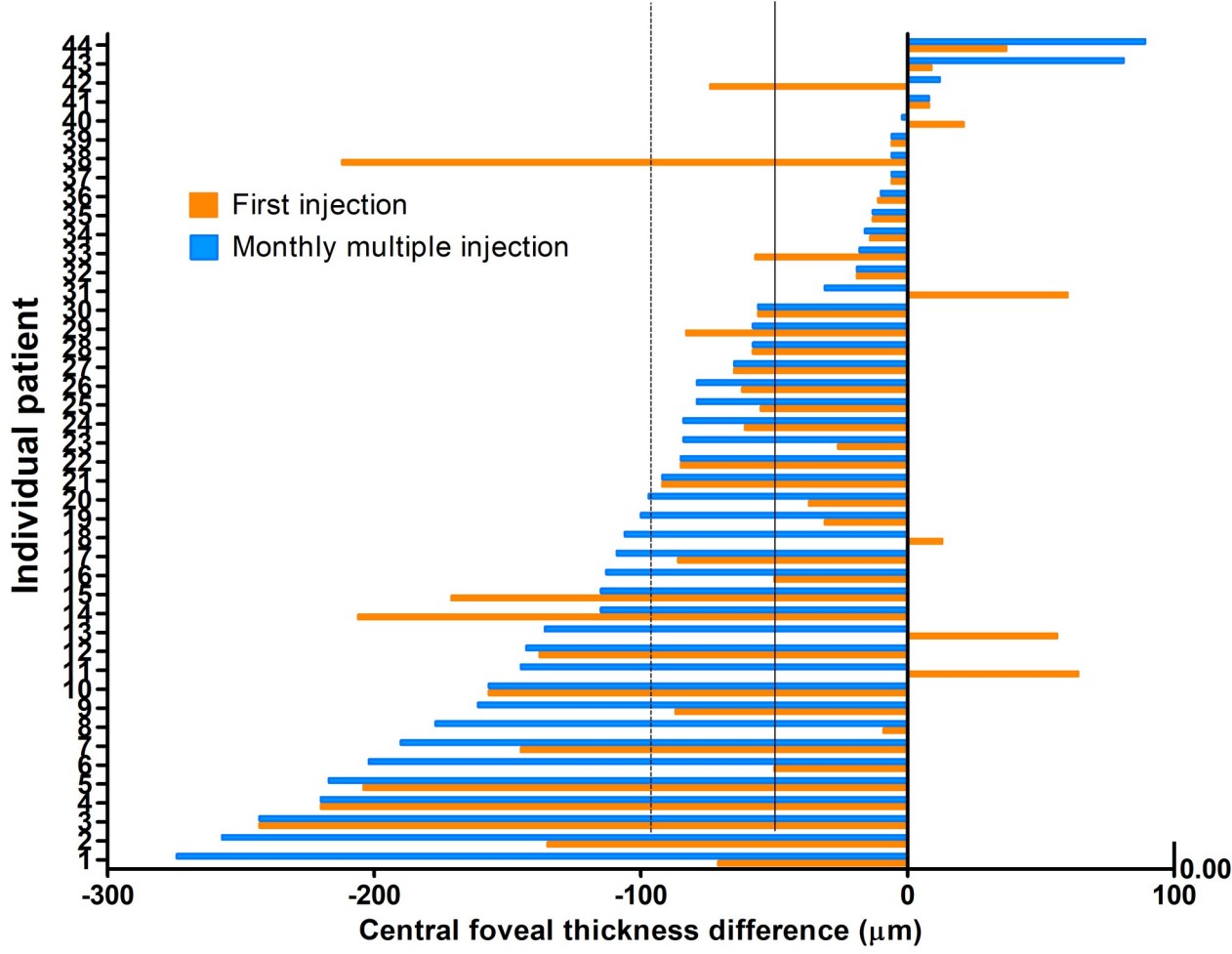

**Fig 4. Central foveal thickness (CFT) difference of each eye.** Solid and dotted line represents median CFT at 1 month after first injection and monthly multiple injections, respectively.

Vitreous is composed of collagen and glycosaminoglycan [26]. It is thought to act as a barrier to diffusion because of its regularly aligned structure, but Knudsen et al showed that fluorescein clearance did not change after vitrectomy in animal eyes [13,27]. It is now controversial whether the clearance of intravitreal anti-VEGF agents is increased in vitrectomized eyes. Two studies of bevacizumab and ranibizumab after vitrectomy reported that the half-lives of these drugs did not differ from those without vitrectomy in rabbit eyes [13,14]. However, these studies were limited as complete PVD was impossible and the remnant vitreous shell could have affected the pharmacokinetics of the anti-VEGF agents. In contrast,

**Table 3. Central foveal thickness (CFT) difference by choroidal neovascularization (CNV) type.**

| CNV type | CFT difference (first injection) | P value | CFT difference (monthly multiple injection) | P value |
|---|---|---|---|---|
| Type 1 (including PCV) (N = 34) | -57.68 ± 74.69 μm | 0.748[a] | -84.18 ± 82.14 μm | 0.688[a] |
| Type 2 (N = 6) | -87.50 ± 94.96 μm | | -81.33 ± 81.49 μm | |
| Type 3 (N = 2) | -40.33 ± 29.14 μm | | -128.00 ± 133.23 μm | |

[a]One-way analysis of variance.

**Table 4. Central foveal thickness (CFT) difference by anti-vascular endothelial growth factor (VEGF) agent type.**

| Type of anti-VEGF agent | CFT difference (first injection) | P value | CFT difference (monthly multiple injection) | P value |
|---|---|---|---|---|
| Bevacizumab (N = 19) | -63.77 ± 20.57 μm | 0.739[a] | -96.89 ± 21.90 μm | 0.230[a] |
| Ranibizumab (N = 13) | -52.77 ± 14.30 μm | | -57.31 ± 19.94 μm | |
| Aflibercept (N = 12) | -77.50 ± 24.80 μm | | -114.00 ± 22.44 μm | |

[a]One-way analysis of variance.

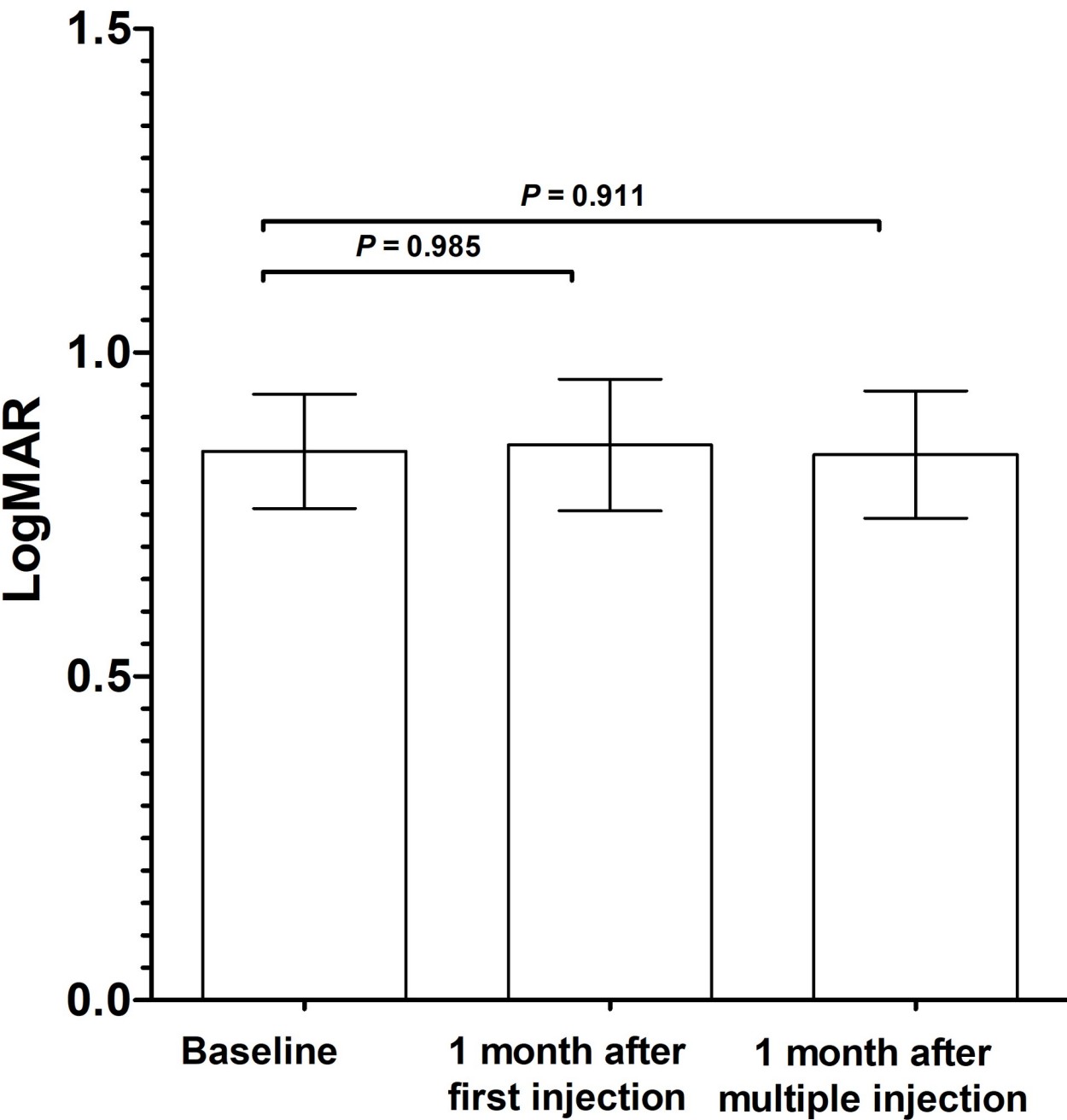

**Fig 5. Change in best-corrected visual acuity (BCVA).** The bars represent mean LogMAR BCVA and the error bars represent standard error of mean.

**Table 5. Comparison of central foveal thickness (CFT) and logarithm of the minimum angle of resolution (LogMAR) best-corrected visual acuity (BCVA) between eyes with and without posterior capsule as well as treatment-naïve and non-treatment-naïve eyes.**

| | | Baseline | 1 month after the first injection | 1 month after monthly multiple injections |
|---|---|---|---|---|
| With posterior capsule (N = 38) | CFT (mean ± SD) | 481.82 ± 147.82 μm | 413.45 ± 135.99 μm (85.8% of baseline) (P < 0.001[a]) | 384.68 ± 140.31 μm (79.8% of baseline) (P<0.001[a]) |
| | LogMAR BCVA[c] (mean ± SD) | 0.89 ± 0.57 | 0.86 ± 0.64 (P = 0.714[a]) | 0.88 ± 0.64 (P = 0.942[a]) |
| Without posterior capsule (N = 6) | CFT (mean ± SD) | 457.50 ± 222.45 μm | 419.33 ± 200.80 μm (91.6% of baseline) (P = 0.917[b]) | 399.83 ± 161.67 μm (87.4% of baseline) (P = 0.249[b]) |
| | LogMAR BCVA (mean ± SD) | 0.62 ± 0.55 | 0.83 ± 0.56 (P = 0.141[b]) | 0.61 ± 0.59 (P = 0.785[b]) |
| Treatment-naïve (N = 17) | CFT (mean ± SD) | 537.88 ± 161.33 μm | 452.41 ± 161.67 μm (84.1% of baseline) (P < 0.001[a]) | 420.18 ± 143.54 μm (78.1% of baseline) (P<0.001[a]) |
| | LogMAR BCVA[d] (mean ± SD) | 0.70 ± 0.59 | 0.63 ± 0.57 (P = 0.181[a]) | 0.60 ± 0.51 (P = 0.071[a]) |
| Non-treatment-naïve (N = 27) | CFT (mean ± SD) | 441.11 ± 144.75 μm | 390.22 ± 128.22 μm (88.5% of baseline) (P = 0.002[a]) | 365.70 ± 138.63 μm (82.9% of baseline) (P < 0.001[a]) |
| | LogMAR BCVA[d] (mean ± SD) | 0.94 ± 0.55 | 0.99 ± 0.64 (P = 0.384[a]) | 0.99 ± 0.67 (P = 0.423[a]) |

[a]Paired t-test.

[b]Wilcoxon signed-rank test.

[c]Two cases without best-corrected visual acuities were excluded.

[d]One case without best-corrected visual acuity was excluded.

another two studies of bevacizumab, ranibizumab, and aflibercept after vitrectomy with lensectomy reported that the half-lives of these drugs were decreased compared to those in non-vitrectomized macaque eyes [16,28]. Combined lensectomy could significantly affect intravitreal clearance because it combines the vitreous cavity and anterior chamber into a single compartment; in particular, drug clearance through the anterior chamber could be altered after combined lensectomy and vitrectomy [13,14]. Christoforidis et al showed that the half-lives of ranibizumab and bevacizumab decreased after vitrectomy without lensectomy in rabbit eyes using a radiolabeled agent, but this study was limited since it did not determine the effect of uncoupling between iodine-124 and anti-VEGF agents [13–15]. In our study, eyes without posterior capsule showed less CFT improvement than those with posterior capsule. This could be explained by the results of the studies mentioned above, since the posterior capsule might prevent anti-VEGF agents from moving to the anterior chamber. Eyes without posterior capsule, such as pseudophakic eyes with sulcus-fixated or scleral-fixated intraocular lenses, had a disrupted or no posterior capsule through which undisturbed fluidic movement was possible between the anterior and posterior chambers. The tight attachment between the peripheral posterior capsule and the intraocular lens could act as a good barrier in the group with posterior capsule despite posterior capsulotomy. The absence of posterior capsule could affect drug clearance since anti-VEGF could be eliminated through the trabecular meshwork. However, further studies with larger numbers of patients are needed to determine its pharmacokinetics and the optimal interval of intravitreal injections.

In contrast to other animal studies, the strength of this human study was that all eyes underwent complete PVD induction with sufficient peripheral vitrectomy with intact posterior capsule in most cases (38 eyes [86.4%]). This enabled almost complete vitrectomy and avoided the effect of remnant vitreous on the pharmacokinetics of intravitreal anti-VEGF agents. In

contrast, in animal eyes, complete vitrectomy without lensectomy was impossible due to the large crystal lens and thick adhesion between the vitreous cortex and retinal surface [13,14,16,28,29]. Thus, through our study, the effect of the vitreous on the anti-VEGF agents could be confirmed as they were efficacious in vitrectomized eyes with posterior capsule. However, it is unclear how long anti-VEGF agents can remain in the vitreous cavity and how effective they can be in human vitrectomized eyes.

Representative studies of bevacizumab, ranibizumab, and aflibercept in non-vitrectomized eyes for treatment-naïve nAMD showed that the degree of reduction in central retinal thickness at 1 month were about 90 μm, about 60 μm, and 156.8 μm, respectively [4,6,30]. The expected degree of reduction in CFT was about 99.35 μm at 1 month after the first injection using the weighted arithmetic mean in previous studies, but the actual reduction was lower in our study [4,6,30]. The mean reduction in CFT, a marker of the anatomical response, was 64.25 ± 78.00 μm at 1 month after the first anti-VEGF injection. The reason for the discrepancy between our and prior studies might be in the prior treatment history and the anti-VEGF agents used. In eyes that received multiple injections, further reductions in CFT (91.75 ± 85.68 μm) were observed. Thus, additional treatment might be beneficial in vitrectomized eyes that are responsive to initial anti-VEGF injections.

In contrast to the effective anatomical outcome, mean BCVAs of eyes were not significantly improved. In a recent real-world study of anti-VEGF with ranibizumab treatment, the mean VA change was -0.062 ± 0.33 logMAR at 1 year from baseline in treatment-naïve patients with a mean of 5.0 ± 2.7 injections [31]. In our study, the mean BCVA change was -0.01 ± 0.37 logMAR at 1 year from baseline with a mean of 4.14 ± 2.12 injections. Most eyes maintained their BCVA, while only one-third showed slight post-treatment BCVA improvements. This result may be attributed to 27 eyes (61.4%) having a prior treatment history and less potential for visual improvement than treatment-naïve eyes [32]. However, in our study, there was no significant change in BCVA after treatment in treatment-naïve eyes (n = 17). This result may be attributed to 5 eyes (29.4%) with breakthrough vitreous hemorrhage from subretinal hemorrhage, which is related with a poor visual prognosis.

Jung et al reported the 2-year result of anti-VEGF injection in 4 vitrectomized nAMD eyes in 4 patients; 3 of 4 patients showed clinical and anatomical improvement, while the other developed atrophy in the macula [18]. Kondo et al studied the efficacy of bevacizumab in vitrectomized eyes with macular edema induced by diabetic retinopathy and branch retinal vein occlusion [21]. In this study, foveal thickness reduced at 1 week but returned to baseline at 1 month after the intravitreal bevacizumab injection [21]. However, since diabetic macular edema and macular edema from retinal vein occlusion have different mechanism from nAMD and are caused by both VEGF and inflammatory cytokines, the result might not exactly correlate with the effect of anti-VEGF agents in vitrectomized eyes with nAMD [33].

Our study findings suggest that the intravitreal concentration of anti-VEGF agents might be sufficiently high and lasting to be effective for nAMD in vitrectomized eyes. Therefore, the use of routine intravitreal anti-VEGF agents can also be recommended in nAMD eyes previously subjected to vitrectomy.

There are several limitations to our study. First, this study was retrospective and evaluated only short-term efficacy in a small number of patients. However, since data on vitrectomized eyes are lacking, our study provides important evidence of the efficacy of intravitreal anti-VEGF agents in vitrectomized nAMD eyes. Second, it was difficult to determine the comparative efficacy of each anti-VEGF agent in vitrectomized eyes since they were used in a non-standardized fashion unlike in prospective clinical trials [4,6,33]. It was also difficult to design a comparative group since several patients in our study had retinal pathologies such as epiretinal membrane or macular hole in addition to nAMD. Third, the duration of efficacy could not be

determined since this study was not designed to determine optimal injection intervals. Further prospective clinical studies are needed to identify the intraocular half-lives and duration of action of anti-VEGF agents in vitrectomized eyes. In addition, since the group without posterior capsule had a very small number of patients (n = 6), a direct comparison between the groups with and without posterior capsule was not possible due to low statistical power. Despite these limitations, this is the first retrospective cohort study to show the effect of intravitreal anti-VEGF agents after complete PVD induction and sufficient peripheral vitrectomy in human eyes.

In conclusion, intravitreal anti-VEGF agents showed favorable anatomical improvement and visual maintenance in eyes with nAMD after PPV. Further studies are necessary to evaluate their long-term efficacies and treatment intervals in vitrectomized eyes.

## Acknowledgments

The authors thank the Medical Research Collaboration Center (MRCC) of Seoul National University Bundang Hospital for assistance in statistical analysis.

## Author Contributions

**Conceptualization:** Se Joon Woo.

**Data curation:** Yongseok Mun.

**Formal analysis:** Yongseok Mun.

**Writing – original draft:** Yongseok Mun.

**Writing – review & editing:** Kyu Hyung Park, Sang Jun Park, Se Joon Woo.

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
