## [Decision Letter · Decision Letter 0]

5 Aug 2020

PONE-D-20-07419

Efficacy of anti-vascular endothelial growth factor agents for treating neovascular age-related macular degeneration in vitrectomized eyes

PLOS ONE

Dear Dr. Woo,

Thank you for submitting your manuscript to PLOS ONE. After careful consideration, we feel that it has merit but does not fully meet PLOS ONE’s publication criteria as it currently stands. Therefore, we invite you to submit a revised version of the manuscript that addresses the points raised during the review process.

We look forward to receiving your revised manuscript.

Kind regards,

Simone Donati, M.D.

Academic Editor

PLOS ONE

Journal Requirements:

Reviewers' comments:

Reviewer's Responses to Questions

**Comments to the Author**

1. Is the manuscript technically sound, and do the data support the conclusions?

Reviewer #1: Partly

2. Has the statistical analysis been performed appropriately and rigorously? 

Reviewer #1: Yes

3. Have the authors made all data underlying the findings in their manuscript fully available?

Reviewer #1: Yes

4. Is the manuscript presented in an intelligible fashion and written in standard English?

Reviewer #1: Yes

5. Review Comments to the Author

Reviewer #1: Number: PONE-D-20-07419:

A small amount of patients and eyes but still clinically a very interesting retrospective original work.

A control group of non-vitrectomized eyes would strengthen this work.

For review comments see below.

Abstract:

Line 18: intravitreal anti-vascular endothelial growth factor is shortened as (VEGF). Shouldn`t it be (anti-VEGF)?

Line 27: Do you have proof for this no A-V communication in group A. In table 1 31 eyes in group A are pseudophakic. Are you shure that there has not been any capsule breaks or rupture in these cases?

Line 31, 35: “multiple injections” – how many injections for how many months or years?

Line 38-40: Please reformulate the conclusion.

Introduction:

Line 48, please add “used”: …are anti-VEGF agents “used” for the treatment of nAMD.

Line line 50: … elderly patients “often” receive vitrectomy… Do you have proof for this statement? How often is vitrectomy actually done?

Line 57: How about clearance? Could you comment on this?

Line 59-63: There have been clinical studies of macular edema due to diabetic retinopathy with a larger amount of eyes, 25 eyes in 2015 and 33 eyes in 2017. Could you comment on this? How can you be sure that all your patients have underwent total vitrectomy and not core vitrectomy? Is this essential for your statement of A-V communication? Is this essential for your conclusion?

Materials and methods:

Line 69-70: In which years did you collect data? Where all patients treatment naïve nAMD before vitrectomy?

Line 76: How many injections for how many months or years? Had any of your patients received laser before anti-VEGF treatment?

Line 79: (3) Please explain this sentence. Is the purpose of anti-VEGF not to dry the macula?

Line 83-84: Do you have proof for this A-V communication?

Line 85: “Before treatment..” – which treatment vitrectomy or anti-VEGF treatment?

Line 90: Which VA did you use, the best measured VA or an average or?

Line 95: When was the first anti-VEGF injection given? Did you always use monthly injections, or PRN injections or?

Line 108: “… three groups…” Weren`t there only 2 groups, group A and B?

Results:

Table 1: Please comment in your discussion on the amount of patients having diabetes mellitus and cancer. Could this influence your results? Did you switch between different types of anti-VEGF?

Figure 1: N=141? On line 69 under “Materials and Methods” you have reviewed 207 medical records? In the same figure you state that “Group A: With A-V communication” and Goup B: Without A-V communication? Isn`t it the opposite? Why did you exclude N=11 eyes receiving anti-VEGF for maintenance?

Discussion:

It could improve the work having a control group. Could you comment on that?

Line 245: What would be the normal improvement of vision in non-vitrectomized eyes having nAMD? Did your vitrectomized patients receive more injection than in general?

6. PLOS authors have the option to publish the peer review history of their article (what does this mean?). If published, this will include your full peer review and any attached files.

Reviewer #1: No

---

## [Author Response · Author response to Decision Letter 0]

9 Sep 2020

September 8, 2020

Simone Donati, M.D.

Academic Editor

PLOS ONE

Dear Dr. Donati,

Thank you for inviting us to submit a revised draft of our manuscript entitled, "Efficacy of anti-vascular endothelial growth factor agents for treating neovascular age-related macular degeneration in vitrectomized eyes" to PLOS ONE. We also appreciate the time and effort you and the reviewer dedicated to providing insightful feedback about strengthening. Thus, it is with great pleasure that we resubmit our article for further consideration. We have incorporated changes that reflect the detailed suggestions you have graciously provided. We hope that our edits and the responses we provide below satisfactorily address all the issues and concerns you and the reviewer noted.

To facilitate review of our revisions, here we provide point-by-point response to the questions and comments delivered in your letter dated Aug 05, 2020.

Reviewer #1: Number: PONE-D-20-07419:

A small amount of patients and eyes but still clinically a very interesting retrospective original work.

A control group of non-vitrectomized eyes would strengthen this work.

For review comments see below.

: Thank you for providing these insights. We also thought it would be better to select a comparative group. However, it was difficult to design a comparative (non-vitrectomized) group with equivalent characteristics since the patients in our study had retinal pathologies in addition to nAMD. We added this point as a limitation (p. 14, lines 279–281). We also supplemented the lack of a comparative group by adding the results of real-world studies (p.13, lines 254–256).

(p. 14, lines 279–281) It was difficult to design a comparative group since several patients in our study had retinal pathologies such as an epiretinal membrane or macular hole in addition to nAMD. 

(p.13, lines 254–256) In a recent real-world study of anti-VEGF with ranibizumab treatment, the mean VA change was -0.062 ± 0.33 logMAR at 1 year from baseline in treatment-naïve patients with a mean of 5.0 ± 2.7 injections.

Abstract:

Line 18: intravitreal anti-vascular endothelial growth factor is shortened as (VEGF). Shouldn`t it be (anti-VEGF)?

: Since we abbreviated only “vascular endothelial growth factor” and not “anti”, we wrote only VEGF in parentheses. But it can be confusing as your you have mentioned in your comment; therefore, we have replaced the term “VEGF” with "anti-VEGF" (p. 2, lines 18–20).

(p. 2, lines 18–20) Evaluate the efficacy of intravitreal anti-vascular endothelial growth factor (anti-VEGF) agents for treatment of neovascular age-related macular degeneration (nAMD) in vitrectomized eyes.

Line 27: Do you have proof for this no A-V communication in group A. In table 1 31 eyes in group A are pseudophakic. Are you sure that there has not been any capsule breaks or rupture in these cases?

: Several patients in group A underwent Nd:YAG laser posterior capsulotomy. However, in this case, there was a little gap between the remaining posterior capsule and the intraocular lens (IOL) that could still could act as aqueous-vitreous barrier. Despite this, drug diffusion and a lack of A-V communication could not be completely prevented. Therefore, instead of using the term “with or without A-V communication,” we directly described the presence or absence of the posterior capsule (i.e., with or without posterior capsule). We also now show the overall result without separating the patients into two groups and then show the subgroup analysis results according to presence or absence of the posterior capsule. We incorporated this throughout our paper.

(p. 2, lines 26–29) A total of 44 eyes from 44 patients were included. The mean central foveal thickness (CFT) was 478.50 ± 156.93 μm at baseline, 414.25 ± 143.55 μm (86.6% of baseline) at 1 month after first injection (P < 0.001), and 386.75 ± 141.45 μm (80.8% of baseline) after monthly multiple injections (2.30 ± 1.07; range, 1–5) (P < 0.001).

(p. 2, lines 32–36) In the subgroup analysis, the CFT of eyes with and without posterior capsule decreased at 1 month after both the first injection (85.8% and 91.6% of baseline, respectively) and monthly multiple injections (79.8% and 87.4% of baseline, respectively). The CFT decreased significantly more in eyes with posterior capsule (P < 0.001).

Line 31, 35: “multiple injections” – how many injections for how many months or years?

: Thank you for raising this important query. Patients with nAMD underwent a mean 2.30 intravitreal anti-VEGF injections (range, 1–5). The mean, standard deviation, and range are described in parentheses (p. 2, lines 26–29). Since monthly means every month, the injection period was 1–5 months. This is also described as “Number of monthly multiple injections” in Table 1.

(p. 2, lines 26–29) The mean central foveal thickness (CFT) was 478.50 ± 156.93 μm at baseline, 414.25 ± 143.55 μm (86.6% of baseline) at 1 month after the first injection (P < 0.001), and 386.75 ± 141.45 μm (80.8% of baseline) after monthly multiple injections (2.30 ± 1.07; range, 1–5) (P < 0.001).

Line 38-40: Please reformulate the conclusion.

: We agree with you and have revised the conclusion accordingly (p. 2, lines 38–39).

(p. 2, lines 38–39) Monthly injections of intravitreal anti-VEGF agents induced favorable anatomical improvement and vision maintenance in vitrectomized eyes with nAMD.

Introduction:

Line 48, please add “used”: …are anti-VEGF agents “used” for the treatment of nAMD.

: We have incorporated your comments by adding “used” accordingly (p. 3, line 47).

(p. 3, line 47) … anti-VEGF agents used for the treatment of nAMD, for which …

Line line 50: … elderly patients “often” receive vitrectomy… Do you have proof for this statement? How often is vitrectomy actually done?

: The retinal diseases that require vitrectomy, such as retinal detachment, vitreous hemorrhage, complicated cataract surgery, macular hole, or macular epiretinal membrane, are usually more prevalent in elderly than in young people. We have added new references and revised the text accordingly (p. 3, lines 49–51).

(p. 3, lines 49–51) In real-world clinical practice, the retinal diseases that require vitrectomy, such as retinal detachment, vitreous hemorrhage, complicated cataract surgery, macular hole, or macular epiretinal membrane, are usually more prevalent in elderly than in young people.[7-12]

Line 57: How about clearance? Could you comment on this?

: Intraocular half-lives of anti-VEGF depended on elimination (clearance), rather than degradation. We have reflected on your comment by revising this sentence (p. 3, line 56). Clearance was also covered in the discussion (p. 12, lines 211–225).

(p. 3, line 56) …retinal specialists argue that the intraocular clearance of intravitreal anti-VEGF would increase.…

(p. 12, lines 211–225) It is now controversial whether the clearance of intravitreal anti-VEGF agents is increased in vitrectomized eyes. Two studies of bevacizumab and ranibizumab treatment after vitrectomy in rabbit eyes reported that the half-lives of these drugs did not differ from those without vitrectomy.[13,14] However, these studies were limited since complete PVD was impossible and the remnant vitreous shell could have affected the pharmacokinetics of the anti-VEGF agents. In contrast, another two studies of bevacizumab, ranibizumab, and aflibercept treatment after vitrectomy with lensectomy in macaque eyes reported that the half-lives of these drugs were decreased compared to those in non-vitrectomized eyes.[16,28] The combined lensectomy could significantly affect intravitreal clearance because it combines the vitreous cavity and anterior chamber into a single compartment; in particular, drug clearance through the anterior chamber could be altered after combined lensectomy and vitrectomy.[13,14] Christoforidis et al showed that the half-lives of ranibizumab and bevacizumab decreased after vitrectomy without a lensectomy in rabbit eyes using a radiolabeled agent, but this study was limited since it did not determine the effect of uncoupling between iodine-124 and anti-VEGF agents.[13-15]

Line 59-63: There have been clinical studies of macular edema due to diabetic retinopathy with a larger amount of eyes, 25 eyes in 2015 and 33 eyes in 2017. Could you comment on this?

: Thank you for your suggestion. Some of these papers include:

1) Intravitreal ranibizumab for diabetic macular oedema in previously vitrectomized eyes (Laugesen CS, 2017)

2) Diabetic Retinopathy Clinical Research Network. Ranibizumab plus prompt or deferred laser for diabetic macular edema in eyes with vitrectomy before anti-vascular endothelial growth factory therapy (Bressler SB, 2015)

We have reflected on your comment by adding these two papers to the reference list and revising the sentence accordingly (p. 3, lines 59–63).

(p. 3, lines 59–63) Several clinical studies reported on the effect of intravitreal anti-VEGF in vitrectomized eyes, but these were only small case series (nAMD, 4 eyes), cohorts of patients with pathologies other than nAMD (macular edema due to diabetic retinopathy or retinal vein occlusion), or cases of patients who underwent core, rather than total vitrectomy.[18-23]

How can you be sure that all your patients have underwent total vitrectomy and not core vitrectomy? 

: We always induce posterior vitreous detachment (PVD) during vitrectomy and remove as much vitreous as possible so the vitreous cortex acts as a barrier to drug clearance. We have incorporated your comment by adding this point accordingly (p. 4, line 92).

(p. 4, line 92) The vitreous cortex was removed as much as possible.

Is this essential for your statement of A-V communication?

: Total vitrectomy is not essential for A-V communication because it is related to the presence of the aqueous-vitreous barrier (i.e., posterior capsule) regardless of vitrectomy. Instead of using the term “A-V communication,” we have directly described the presence or absence of the posterior capsule (with or without the posterior capsule) as mentioned above (question for line 27).

Is this essential for your conclusion?

: Total vitrectomy is essential to our conclusion since the remnant vitreous cortex interferes with evaluation of the vitreous clearance. We have clarified that point (p. 12, lines 215–216; p. 15, lines 284–286).

(p. 12, lines 215–216) However, these studies were limited as complete PVD was impossible and the remnant vitreous shell could have affected the pharmacokinetics of the anti-VEGF agents.

(p. 14-15, lines 284–286) Despite these limitations, this is the first retrospective cohort study to show the effect of intravitreal anti-VEGF agents after complete PVD induction and sufficient peripheral vitrectomy in human eyes.

Materials and methods:

Line 69-70: In which years did you collect data?

: Thank you for noting this. We collected data from March 2009 to December 2018. We have incorporated your comment by adding the data collection period accordingly (p. 4, lines 69–70).

(p. 4, lines 69–70) We retrospectively reviewed the electronic medical records of 141 vitrectomized eyes of 141 patients with nAMD treated between March 2009 and December 2018. 

Where all patients treatment naïve nAMD before vitrectomy?

: As shown in Table 1, 17 patients were treatment-naïve (p. 7, Table 1).

Line 76: How many injections for how many months or years?

: We did not limit the number of intravitreal injections or duration of treatment. Patients received intravitreal anti-VEGF injections repeatedly until the macular fluid disappeared regardless of the number of injections. We have revised the sentence for clarity accordingly (p. 5, lines 98–101).

(p. 5, lines 98–101) Based on OCT findings, patients who showed an insufficient response were given monthly multiple injections until a sufficient decrease in intraretinal or subretinal fluid – i.e. fluid-free macula – was achieved. 

Had any of your patients received laser before anti-VEGF treatment?

: Two patients received photodynamic therapy (PDT) once each before enrolling in our study. No patients received PDT during the study period. We reflected your comment by adding this point to the Results section (p. 6, lines 130–131).

(p. 6, lines 130–131) 2 eyes (4.5%) received photodynamic therapy before enrolling in our study. 

Line 79: (3) Please explain this sentence. Is the purpose of anti-VEGF not to dry the macula?

: Thank you for putting forward an interesting question. We were not interested in patients with proactive treatment since we focused on whether the activity (macular fluid) disappeared due to the anti-VEGF injection. We have revised the sentence accordingly (p. 4, lines 80–81).

(p. 4, lines 80–81) (3) no anti-VEGF injections despite macular fluid – i.e., proactive treatment; 

Line 83-84: Do you have proof for this A-V communication?

: Thank you for pointing that out. “A-V communication” means that there was enough space through which the aqueous and vitreous humor could diffuse into each other. We thought that the presence or absence of posterior capsule represented this point. However, even with the posterior capsule acting as an aqueous-vitreous barrier, drug diffusion and a lack of A-V communication could not be completely prevented. Therefore, instead of using the term “with or without A-V communication,” we directly described the presence or absence of the posterior capsule. (i.e., with or without posterior capsule). We also changed the text to show the overall result without separating the patients into two groups and then showed the subgroup analysis results according to the presence or absence of the posterior capsule. (p. 5, lines 108–109)

(p. 5, lines 108–109) A subgroup analysis was performed by comparing patients with versus those without posterior capsule.

Line 85: “Before treatment..” – which treatment vitrectomy or anti-VEGF treatment?

: This refers to the first anti-VEGF treatment after vitrectomy. We have reflected on your comment by revising this sentence for clarity (p. 4, line 83).

(p. 4, line 83) Before the first anti-VEGF injection after vitrectomy, …

Line 90: Which VA did you use, the best measured VA or an average or?

: We did not use average visual acuity or the maximum visual acuity measured during the study period. In most cases, we used the best corrected visual acuity immediately before treatment and the visual acuity at 1 month after treatment. We have reflected your comment by adding this point (p. 5, lines 104–106; p. 10, lines 185–185).

(p. 5, lines 104–106) VA was collected with the CFT and converted to the logarithm of the minimum angle of resolution (logMAR) before the analysis. 

(p. 10, lines 185–186) Uncorrected VA was used in only 2 cases since the best corrected VA was not measured.

Line 95: When was the first anti-VEGF injection given?

: The first anti-VEGF injection was administered to a treatment-naïve patient within 2 weeks after the diagnosis of nAMD. The first anti-VEGF injection was administered to a non-treatment-naïve patient within 2 weeks after macular fluid was observed. We have added a new sentence to clarify this point (p. 5, lines 94–97).

(p. 5, lines 94–97) The first anti-VEGF injection was administered to a treatment-naïve patient within 2 weeks after the diagnosis of nAMD. The first anti-VEGF injection was administered to a non-treatment-naïve patient within 2 weeks after macular fluid was observed. 

Did you always use monthly injections, or PRN injections or?

: As described in lines 98–101, if the macular fluid was not dried up at 1 month after the injection, an additional injection was given and the examination repeated 1 month later. Therefore, it could be said that the PRN (reactive) strategy was conducted.

(p. 5, lines 98–101) Based on OCT findings, patients who showed an insufficient response were given monthly multiple injections until a sufficient decrease in intraretinal or subretinal fluid – i.e. fluid-free macula – was achieved.

Line 108: “… three groups…” Weren`t there only 2 groups, group A and B?

: Analysis of variance (ANOVA) was used to compare the three subgroups. The P values in Table 3 (three groups according to choroidal neovascularization type) and Table 4 (three groups according to anti-VEGF agent type) were calculated using ANOVA.

Results:

Table 1: Please comment in your discussion on the amount of patients having diabetes mellitus and cancer. Could this influence your results?

: The presence of diabetes mellitus or cancer did not affect anti-VEGF use.

Did you switch between different types of anti-VEGF?

: Only 2 patients received bevacizumab and ranibizumab due to drug switching. The other 42 patients received only one type of anti-VEGF agent. One patient received four monthly injections (1 of ranibizumab, 3 of bevacizumab), while the other received five monthly injections (3 of ranibizumab, 2 of bevacizumab). However, this switching was not due to the ineffectiveness of ranibizumab; rather, ranibizumab was not covered by insurance since their visual acuities after treatment did not fulfill the reimbursement criteria. Therefore, the drug was switched to bevacizumab, which could be used regardless of insurance coverage.

Figure 1: N=141? On line 69 under “Materials and Methods” you have reviewed 207 medical records? 

: Thank you for identifying this problem. The total of N = 141 is correct. We agree with you and have reflected on your comment by correcting this number in the Materials and Methods section (p. 3, line 69–70).

(p. 3, line 69–70) We retrospectively reviewed the electronic medical records of 141 vitrectomized eyes of 141 patients with nAMD treated between March 2009 and December 2018. 

In the same figure you state that “Group A: With A-V communication” and Group B: Without A-V communication? Isn`t it the opposite?

: Thank you for pointing out the error. The expression about the A-V communication of Groups A and B was reversed. As mentioned in the answer to the question for line 27, instead of using the term “with or without A-V communication”, we directly described the presence or absence of the posterior capsule (i.e., with or without posterior capsule). We also changed the text to show the overall result without separating the patients into two groups first and then show the subgroup analysis results according to posterior capsule status. We incorporated this in Figure 1.

Why did you exclude N=11 eyes receiving anti-VEGF for maintenance?

: As mentioned above, we excluded 11 eyes that received proactive anti-VEGF treatment since we could not directly evaluate the treatment response after the anti-VEGF injections. Please see the answer to the question for line 79.

Discussion:

It could improve the work having a control group. Could you comment on that?

: Thank you for providing these insights. Please see our answer to the first question.

Line 245: What would be the normal improvement of vision in non-vitrectomized eyes having nAMD? 

: That is a very important perspective. We have added the results of real-world studies that enrolled non-vitrectomized eyes to reflect your comment (p. 13, lines 254–256).

(p.13, lines 254–256) In a recent real-world study of anti-VEGF with ranibizumab treatment, the mean VA change was -0.062 ± 0.33 logMAR at 1 year from baseline in treatment-naïve patients with a mean of 5.0 ± 2.7 injections.

Did your vitrectomized patients receive more injection than in general?

: That is a key question. Our study could only suggest that anti-VEGF was effective in vitrectomized eyes. It was difficult to directly compare this result with that in non-vitrectomized eyes due to difficulty designing a comparative group as mentioned above (first question). Instead, we added the mean number of injections in 1 year and the VA at 1 year after the first injection to compare our result with that of the other real-world study (p. 6, lines 131–132; p. 10, line 181; p. 13–14, lines 254–258).

(p. 6, lines 131–132) The total mean anti-VEGF injections was 4.14 ± 2.12 at 1 year after the first injection.

(p. 10, line 181) Mean LogMAR VA was 0.90 ± 0.66 at 1 year after the first injection.

(p. 13–14, lines 254–258) In a recent real-world study of anti-VEGF with ranibizumab treatment, the mean VA change was -0.062 ± 0.33 logMAR at 1 year from baseline in treatment-naïve patients with a mean of 5.0 ± 2.7 injections.[31] In our study, the mean VA change was -0.0090 ± 0.36 logMAR at 1 year from baseline with a mean 4.14 ± 2.12 injections. Most eyes maintained their VA, while only one-third showed slight post-treatment VA improvements. 

We also modified Figures 3 and 5 to bar graphs to help readers better understand our results.

Thank you again for the opportunity to strengthen our manuscript with your valuable comments and queries. We have worked hard to incorporate your feedback and hope that these revisions persuade you to accept our submission.

---

## [Decision Letter · Decision Letter 1]

25 Jan 2021

PONE-D-20-07419R1

Efficacy of anti-vascular endothelial growth factor agents for treating neovascular age-related macular degeneration in vitrectomized eyes

PLOS ONE

Dear Dr. Woo,

Thank you for submitting your manuscript to PLOS ONE. After careful consideration, we feel that it has merit but does not fully meet PLOS ONE’s publication criteria as it currently stands. Therefore, we invite you to submit a revised version of the manuscript that addresses the points raised during the review process.

We look forward to receiving your revised manuscript.

Kind regards,

Manuel Alberto de Almeida e Sousa Falcão, M.D., Ph.D.

Academic Editor

PLOS ONE

Reviewers' comments:

Reviewer's Responses to Questions

**Comments to the Author**

1. If the authors have adequately addressed your comments raised in a previous round of review and you feel that this manuscript is now acceptable for publication, you may indicate that here to bypass the “Comments to the Author” section, enter your conflict of interest statement in the “Confidential to Editor” section, and submit your "Accept" recommendation.

Reviewer #1: (No Response)

Reviewer #2: (No Response)

2. Is the manuscript technically sound, and do the data support the conclusions?

Reviewer #1: Yes

Reviewer #2: Yes

3. Has the statistical analysis been performed appropriately and rigorously? 

Reviewer #1: Yes

Reviewer #2: Yes

4. Have the authors made all data underlying the findings in their manuscript fully available?

Reviewer #1: Yes

Reviewer #2: Yes

5. Is the manuscript presented in an intelligible fashion and written in standard English?

Reviewer #1: Yes

Reviewer #2: Yes

6. Review Comments to the Author

Reviewer #1: Thank You so much for answering all my comments so thoroughly. It has strengthened the manuscript a lot and is now more readable. I have some small comments for this new manuscript.

In line 22: the word using should be exchanged to with. In line 70: there is missing an a in the word treated. In line 100 2 words are written together (untila).

Using the description of with or without posterior capsule is much better than AV communication. Instead of with and without posterior capsule you could consider presence of or absence of. Are your sure that the posterior capsule always is intact when it is present?

The very thorough explanation you are given me as a reviewer about switching between types of anti-VEGF you should consider adding in your manus or in table 1.

Reviewer #2: This is an interesting paper but some points need to be clarified.

It is not clear how many patients were already being treated for wet AMD at the time of the vitrectomy. I think these are the non-naive patients, but this needs to be confirmed. But this is a very important aspect. Non-naive patients are patients that are already being treated for their wet AMD and therefore one would not expect them to improve vision. In wet AMD, patients usually improve vision after the first 2-3 injections so we would not expect them to improve. Unless they were the active after the vitrectomy and this is a completely different situation? Were they being undertreated?

The authors should have analysed the non-naive and naive patients as different subgroups….

Also, we do not know what were the reasons for vitrectomy in both of these groups and this also has implications on vision... were they retinal detachments? macular holes? epirretinal membranes? vitreous haemorrhage from previous injection?

Having a subgroup analysis of just 6 patients should be avoided as it is a very small number of patients and no valid conclusions can be obtained. Maybe they should have been left out or not directly compared.

It is not clear what the authors mean by no posterior capsule? Were these patients aphakic? Were they anterior chamber IOLs? Scleral suture? in the bag IOLs that had previous been treated with capsulotomy?

This needs to be clarified as well.

7. PLOS authors have the option to publish the peer review history of their article (what does this mean?). If published, this will include your full peer review and any attached files.

Reviewer #1: No

Reviewer #2: **Yes: **Manuel Falcão

---

## [Author Response · Author response to Decision Letter 1]

15 Feb 2021

February 15, 2021

Manuel Alberto de Almeida e Sousa Falcão, M.D., Ph.D.

Academic Editor

PLOS ONE

Dear Dr. Manuel Alberto de Almeida e Sousa Falcão,

Thank you for inviting us to submit a revised draft of our manuscript entitled, "Efficacy of anti-vascular endothelial growth factor agents for treating neovascular age-related macular degeneration in vitrectomized eyes" to PLOS ONE. We also appreciate the time and effort you and the reviewer dedicated to providing insightful feedback about strengthening. Thus, it is with great pleasure that we resubmit our article for further consideration. We have incorporated changes that reflect the detailed suggestions you have graciously provided. We hope that our edits and the responses we provide below satisfactorily address all the issues and concerns you and the reviewer noted.

To facilitate review of our revisions, here we provide point-by-point response to the questions and comments delivered in your letter dated Jan 25, 2021.

Reviewer #1

Thank You so much for answering all my comments so thoroughly. It has strengthened the manuscript a lot and is now more readable. I have some small comments for this new manuscript.

-> Thank you for your comment. Changes made in the revised manuscript have been highlighted in yellow.

In line 22: the word using should be exchanged to with.

-> We have changed “using” to “with” accordingly.

(p. 2, lines 22-23) The medical records were reviewed of nAMD patients treated with anti-VEGF agents who previously underwent pars plana vitrectomy (PPV).

In line 70: there is missing an a in the word treated.

-> We have changed “treted” to “treated” accordingly.

(p. 4, lines 69-70) We retrospectively reviewed the electronic medical records of 141 vitrectomized eyes of 141 patients with nAMD treated between March 2009 to December 2018

In line 100: 2 words are written together (untila).

-> We have added a space between “until” and “a” accordingly.

(p. 5, lines 101-104) Based on OCT findings, patients who showed an insufficient response were administered repeated injections monthly until a sufficient decrease in intraretinal or subretinal fluid, i.e., fluid-free macula, was achieved.

Using the description of with or without posterior capsule is much better than AV communication. Instead of with and without posterior capsule you could consider presence of or absence of. Are your sure that the posterior capsule always is intact when it is present?

-> Posterior capsulotomies were performed in some cases in the group with posterior capsule. However, even in these cases, the peripheral posterior capsules were present and firmly attached to the intraocular lenses because the openings were created in the center of the capsules. The tight attachment between the peripheral posterior capsule and the intraocular lens could act as a good barrier between the anterior and posterior chambers. This could prevent anti-VEGF agents from moving to the anterior chamber. In this respect, the posterior capsules were intact in the group with posterior capsule. We have revised the text by adding the answer to your question in the Discussion section.

(p. 13-14, lines 244-246) The tight attachment between the peripheral posterior capsule and the intraocular lens could act as a good barrier in the group with posterior capsule despite posterior capsulotomy.

The very thorough explanation you are given me as a reviewer about switching between types of anti-VEGF you should consider adding in your manus or in table 1.

-> Thank you for raising this important question. We have revised Table 1 by adding a footnote about switching anti-VEGF agents.

(p. 6-8, lines 140-145) Table 1. Demographic and baseline characteristics of the 44 study participants

aOne patient received 3 bevacizumab injections after 1 ranibizumab injection and the other patient received 2 bevacizumab injections after 3 ranibizumab injections due to insurance coverage.

Reviewer #2

This is an interesting paper but some points need to be clarified.

-> Thank you for your thorough review and suggestions. Changes made in the revised manuscript have been highlighted in yellow.

It is not clear how many patients were already being treated for wet AMD at the time of the vitrectomy. I think these are the non-naïve patients, but this needs to be confirmed. But this is a very important aspect.

-> You have raised an important question: Twenty-seven eyes (61.4%) were non-treatment-naïve, as mentioned in lines 133–134.

(p. 6, lines 133-134) Twenty-seven eyes (61.4%) were non-treatment-naïve and 2 eyes (4.5%) received photodynamic therapy before enrolling in our study.

Non-naïve patients are patients that are already being treated for their wet AMD and therefore one would not expect them to improve vision. In wet AMD, patients usually improve vision after the first 2-3 injections so we would not expect them to improve. Unless they were the active after the vitrectomy and this is a completely different situation? Were they being undertreated?

-> Neovascular age-related macular degeneration (nAMD) was treated the same manner in vitrectomized and non-vitrectomized eyes. Anti-vascular endothelial growth (VEGF) injection was initiated when the lesion was active. As mentioned in lines 101–104, the treatment response was monitored by optical coherence tomography (OCT) at 1 month after each injection, and patients who showed an insufficient response were given monthly multiple injections until a sufficient decrease in intraretinal or subretinal fluid, i.e., fluid-free macula, was achieved. Therefore, nAMD in vitrectomized eyes is rarely undertreated. A treat-and-extent (T&E) or pro re nata (PRN) regimen was used after fluid-free macula was achieved. Three loading doses monthly for the first 3 months were administered to treatment-naïve patients. However, there might be discrepancies between our study and randomized control trials based on intensive treatment such as monthly fixed dosing of anti-VEGF as our study used real-world data. We have clarified this point in the revised manuscript.

(p. 5, lines 97-100) The treatment protocol was identical for vitrectomized and non-vitrectomized patients; 3 monthly loading injections were administered initially to treatment-naïve patients. Thereafter, patients received anti-VEGF injections pro re nata or according to a treat-and-extend regimen at the physician’s discretion.

The authors should have analyzed the non-naïve and naive patients as different subgroups….

-> We agree with your assessment and have added results of the subgroup analysis by treatment-naïve status.

(p. 5, lines 110-112) Subgroup analyses were performed by comparing patients with and without posterior capsule as well as treatment-naïve patients and non-treatment-naïve patients.

(p. 10-11, lines 204-210) In the subgroup analysis of the posterior capsule, the mean CFT of eyes with posterior capsule (n = 38) decreased significantly after the first injection (P < 0.001) and monthly multiple injections (P < 0.001). There was no significant change in baseline VA after the first injection (P = 0.714) or monthly multiple injections (P = 0.877). The mean CFT of eyes without posterior capsule (n = 6) did not significantly improve at 1 month after the first injection (P = 0.917) or monthly multiple injections (P= 0.249). There was no significant difference in VA after the first injection (P = 0.141) or monthly multiple injections (P = 0.785) (Table 5).

(p. 11-12, lines 212-216) Table 5. Comparison of central foveal thickness (CFT) and logarithm of the minimum angle of resolution (LogMAR) visual acuity (VA) between patients with and without posterior capsule as well as treatment-naïve patients and non-treatment-naïve patients

(p. 15, lines 278-281) However, in our study, there was no significant change in VA after treatment in treatment-naïve eyes (N = 17). This result may be attributed to 5 eyes (29.4%) with breakthrough vitreous hemorrhage from subretinal hemorrhage, which is related with a poor visual prognosis.

Also, we do not know what were the reasons for vitrectomy in both of these groups and this also has implications on vision... were they retinal detachments? macular holes? epirretinal membranes? vitreous haemorrhage from previous injection?

-> We agree with your comment. We have revised Table 1 by adding treatment-naïve and non-treatment-naïve groups accordingly.

(p. 6-8, lines 140-145) Table 1. Demographic and baseline characteristics of the 44 study participants.

Having a subgroup analysis of just 6 patients should be avoided as it is a very small number of patients and no valid conclusions can be obtained. Maybe they should have been left out or not directly compared.

-> Thank you for pointing this out. We completely agree with you. We have incorporated your comments by adding a statement regarding the difficulty of direct comparison between the two groups in the Discussion section.

(p. 16, lines 305-308) In addition, since the group without posterior capsule had a very small number of patients (n = 6), a direct comparison between the groups with and without posterior capsule was not possible due to low statistical power.

It is not clear what the authors mean by no posterior capsule? Were these patients aphakic? Were they anterior chamber IOLs? Scleral suture? in the bag IOLs that had previous been treated with capsulotomy? This needs to be clarified as well.

-> Thank you for your suggestion. We have revised the text by adding the intraocular lens status accordingly.

(p. 13, lines 241-244) Eyes without posterior capsule, such as pseudophakic eyes with sulcus-fixated or scleral-fixated intraocular lenses, had a disrupted or no posterior capsule through which undisturbed fluidic movement was possible between the anterior and posterior chambers.

Again, thank you for giving us the opportunity to strengthen our manuscript with your valuable comments and queries. We have worked hard to incorporate your feedback and hope that these revisions persuade you to accept our submission.

---

## [Decision Letter · Decision Letter 2]

28 Mar 2021

PONE-D-20-07419R2

Efficacy of anti-vascular endothelial growth factor agents for treating neovascular age-related macular degeneration in vitrectomized eyes

PLOS ONE

Dear Dr. Woo,

Thank you for submitting your manuscript to PLOS ONE. After careful consideration, we feel that it has merit but does not fully meet PLOS ONE’s publication criteria as it currently stands. Therefore, we invite you to submit a revised version of the manuscript that addresses the points raised during the review process.

We look forward to receiving your revised manuscript.

Kind regards,

Manuel Alberto de Almeida e Sousa Falcão, M.D., Ph.D.

Academic Editor

PLOS ONE

Journal Requirements:

Reviewers' comments:

Reviewer's Responses to Questions

**Comments to the Author**

1. If the authors have adequately addressed your comments raised in a previous round of review and you feel that this manuscript is now acceptable for publication, you may indicate that here to bypass the “Comments to the Author” section, enter your conflict of interest statement in the “Confidential to Editor” section, and submit your "Accept" recommendation.

Reviewer #1: All comments have been addressed

Reviewer #2: All comments have been addressed

2. Is the manuscript technically sound, and do the data support the conclusions?

Reviewer #1: Yes

Reviewer #2: Partly

3. Has the statistical analysis been performed appropriately and rigorously? 

Reviewer #1: Yes

Reviewer #2: Yes

4. Have the authors made all data underlying the findings in their manuscript fully available?

Reviewer #1: Yes

Reviewer #2: No

5. Is the manuscript presented in an intelligible fashion and written in standard English?

Reviewer #1: Yes

Reviewer #2: Yes

6. Review Comments to the Author

Reviewer #1: Indeed a thorough and good revision. All my comments have been addressed. It has strenghtened the manus to descriminate between treatment-naïve and non-treatment-naïve eyes. You can consider to add "intact" whenever using "with posterior capsule". You can consider to rewrite the sentence on line 87-88, exchange which with where, add VA and omit the word "together": "In cases where best corrected VA was not measured, uncorrected VA was collected."

Reviewer #2: Thank you for addressing all my comments. The paper has become clearer now but there are still some points that I have now understood and need further clarification.

Firstly, please remove patients with uncorrected visual acuity. I am sorry for not having seen this in the previous review. This is not acceptable. Non corrected visual acuity has a ceiling affect that cannot used in scientific reports.

Secondly, I need to understand when first intravitreal injections were performed for most of the patients and if, in the patients that were already being treated, if anti-VEGF was used at the end of the vitrectomy.

In the 17 eyes that were treatment naive, 12 had had previous vitrectomies for other reasons. It is clear for these patients that visual acuity and OCT were measured after the vitrectomy and just before the first intravitreal treatment. And, probably, these eyes had increases in visual acuity.

However, the remaining five eyes were treated due to a breakthrough vitreous haemorrhage. This means that probably OCT was not possible to do before the vitrectomy. Also, at the time of the vitrectomy, did the surgeon perform an intravitreal anti-VEGF injection at the end of the procedure when he saw that the cause of the vitreous haemorrhage was a macular neovascularization? to avoid another breakthrough vitreous haemorrhage? If so, then the first OCT performed after the vitrectomy was not naive, but after the first injection... and therefore it really is not a baseline evaluation.

27 eyes that were already being treated. This means that the potential for visual recovery would be difficult because AMD patients only improve vision after the first injections. and this is an important reason for the population not having increases in visual acuity overall.

However, 16 eyes had breakthrough vitreous hemorrhages, which means that probably they were being undertreated... once again, were they not anti-VEGF treated during the vitrectomy?

Please correct the abstract where it says " CFT decreased by 91.6% and 87.4% at 1 month after the first injection and after monthly multiple injections, respectively, without statistical significance."

Please state your criteria for improving vision, stable vision or worse vision.

7. PLOS authors have the option to publish the peer review history of their article (what does this mean?). If published, this will include your full peer review and any attached files.

Reviewer #1: No

Reviewer #2: No

---

## [Author Response · Author response to Decision Letter 2]

15 Apr 2021

April 13, 2021

Manuel Alberto de Almeida e Sousa Falcão, M.D., Ph.D.

Academic Editor

PLOS ONE

Dear Dr. Manuel Alberto de Almeida e Sousa Falcão,

Thank you for inviting us to submit a revised draft of our manuscript titled, "Efficacy of anti-vascular endothelial growth factor agents for treating neovascular age-related macular degeneration in vitrectomized eyes" to PLOS ONE. We also appreciate the time and effort you and the reviewer dedicated to providing insightful feedback about strengthening. Thus, it is with great pleasure that we resubmit our article for further consideration. We have incorporated changes that reflect the detailed suggestions you have graciously provided. We hope that our edits and the responses we provide below satisfactorily address all the issues and concerns you and the reviewer noted.

To facilitate review of our revisions, here we provide point-by-point response to the questions and comments delivered in your letter dated March 28, 2021.

Reviewer #1

Indeed a thorough and good revision. All my comments have been addressed. It has strengthened the manus to discriminate between treatment-naïve and non-treatment-naïve eyes.

You can consider to add "intact" whenever using "with posterior capsule".

-> Thank you for your suggestion. We also considered the word “intact” at the time of revision. However, this word can be misunderstood as the absence of defects in the posterior capsule. As mentioned on page 14, line 246, some patients might have undergone posterior capsulotomy although the tight attachment between the peripheral posterior capsule and the intraocular lens guaranteed no change of drug clearance compared to the intact posterior capsules. Therefore, we have decided not to use the word “intact.” Once again, thank you very much your kind suggestion.

You can consider to rewrite the sentence on line 87-88, exchange which with where, add VA and omit the word "together": "In cases where best corrected VA was not measured, uncorrected VA was collected."

-> We have changed the sentence according to your and another reviewer’s suggestions.

(p. 4, lines 89-90) Two cases without best-corrected VAs were excluded from the visual outcome analysis.

Reviewer #2

Reviewer #2: Thank you for addressing all my comments. The paper has become clearer now but there are still some points that I have now understood and need further clarification.

Firstly, please remove patients with uncorrected visual acuity. I am sorry for not having seen this in the previous review. This is not acceptable. Non corrected visual acuity has a ceiling affect that cannot used in scientific reports.

-> Thank you for pointing this out. Owing to its retrospective nature, best-corrected visual acuities (BCVAs) could not be collected in two cases as mentioned in page 10, lines 192 and 193. We have deleted the two cases from the dataset for the visual outcome analysis and performed an analysis on only those cases with BCVA. However, for the analysis of anatomic outcomes, such as central foveal thickness on OCT, all cases were included.

(p. 2, lines 29-32) The mean logarithm of the minimum angle of resolution VA was 0.85 ± 0.578 at baseline, 0.86 ± 0.63 after the first injection, and 0.84 ± 0.64 after monthly multiple injections. VA improved in 39.5% at 1 month after first injection and 45.2% at 1 month after monthly multiple injections.

(p. 4, lines 89-90) Two cases without best-corrected VAs were excluded from the visual outcome analysis.

(p. 11, lines 188-195) In patients with best-corrected VA (n = 42), mean LogMAR VA was 0.85 ± 0.57 at baseline, 0.86 ± 0.63 at 1 month after the first injection, and 0.84 ± 0.64 at 1 month after monthly multiple injections. There was also no significant difference in VA between pre- and post-injection (P = 0.985 and 0.911, respectively). Mean LogMAR VA was 0.83 ± 0.61 at 1 year after the first injection. Of the 38 eyes (those of 4 eyes were missing), VA was improved in 15 eyes (39.5%), maintained in 12 eyes (31.6%), and deteriorated in 11 eyes (28.9%) at 1 month after the first injection. Of the 42 eyes, VA was improved in 19 eyes (45.2%), maintained in 14 eyes (33.3%), and deteriorated in 9 eyes (21.4%) at 1 month after monthly multiple injections.

(p. 11, lines 197-198) Fig 5. Change in visual acuity (VA). The bars represent mean LogMAR VA and the error bars represent standard error of mean.

(p. 11, lines 202-203) There was no significant change in VA (n = 36) after the first injection (P = 0.714) or monthly multiple injections (P = 0.942).

(p. 11, lines 209-210) However, there was no significant change in VA (n = 16) after the first injection (P = 0.181) or monthly multiple injections (P = 0.071).

(p. 11-12, lines 212-213) however, there was no significant change in VA (n = 26) after the first injection (P = 0.384) or monthly multiple injections (P = 0.423) (Table 5).

(p. 11, lines 215-221) Table 5. Comparison of central foveal thickness (CFT) and logarithm of the minimum angle of resolution (LogMAR) visual acuity (VA) between patients with and without posterior capsule as well as treatment-naïve patients and non-treatment-naïve patients

Secondly, I need to understand when first intravitreal injections were performed for most of the patients and if, in the patients that were already being treated, if anti-VEGF was used at the end of the vitrectomy.

In the 17 eyes that were treatment naive, 12 had had previous vitrectomies for other reasons. It is clear for these patients that visual acuity and OCT were measured after the vitrectomy and just before the first intravitreal treatment. And, probably, these eyes had increases in visual acuity.

However, the remaining five eyes were treated due to a breakthrough vitreous hemorrhage. This means that probably OCT was not possible to do before the vitrectomy. Also, at the time of the vitrectomy, did the surgeon perform an intravitreal anti-VEGF injection at the end of the procedure when he saw that the cause of the vitreous hemorrhage was a macular neovascularization? to avoid another breakthrough vitreous hemorrhage? If so, then the first OCT performed after the vitrectomy was not naive, but after the first injection... and therefore it really is not a baseline evaluation.

-> Thank you for raising the important question. In the treatment-naïve group, eyes with breakthrough vitreous hemorrhage did not receive anti-VEGF at the end of vitrectomy; therefore, OCT data at the time of the first injection could be regarded as baseline.

(p. 5, lines 103-104) In the treatment-naïve group, eyes with breakthrough vitreous hemorrhage did not receive anti-VEGF at the end of vitrectomy.

27 eyes that were already being treated. This means that the potential for visual recovery would be difficult because AMD patients only improve vision after the first injections. and this is an important reason for the population not having increases in visual acuity overall.

However, 16 eyes had breakthrough vitreous hemorrhages, which means that probably they were being undertreated... once again, were they not anti-VEGF treated during the vitrectomy?

-> In the non-treatment-naïve group, six eyes with breakthrough vitreous hemorrhage received intravitreal anti-VEGF injections at the end of vitrectomy. However, the cohort entry date of these patients was not the day of vitrectomy but the one after. Since vitreous hemorrhage made it difficult to define the pre-treatment status, such as baseline CFT, they were enrolled at the time of receiving anti-VEGF injections after vitreous hemorrhage had been resolved. We have added this point in the Methods section.

(p. 4, 82-84 lines) Since eyes with breakthrough vitreous hemorrhage (VH) could not be evaluated, non-treatment-naïve eyes with breakthrough VH were enrolled at the time of the first anti-VEGF injection after resolution of VH.

Please correct the abstract where it says " CFT decreased by 91.6% and 87.4% at 1 month after the first injection and after monthly multiple injections, respectively, without statistical significance."

-> Thank you for pointing that out. We have revised the text accordingly.

(p. 2, lines 32-36) In the subgroup analysis, CFT of eyes with the posterior capsule decreased significantly by 85.8% and 79.8% at 1 month after the first injection and after monthly multiple injections, respectively. CFT of eyes without the posterior capsule decreased by 91.6% and 87.4% at 1 month after the first injection and after monthly multiple injections, respectively, without statistical significance.

Please state your criteria for improving vision, stable vision or worse vision.

-> Improvement and worsening were defined as a change ≥0.1 LogMAR from baseline. We have added this criterion in the Methods section.

(p. 5, lines 113-114) Improvement and worsening of VA were defined as a change ≥0.1 LogMAR from baseline.

---

## [Editor Report · Decision Letter 3]

10 May 2021

Efficacy of anti-vascular endothelial growth factor agents for treating neovascular age-related macular degeneration in vitrectomized eyes

PONE-D-20-07419R3

Dear Dr. Woo,

We’re pleased to inform you that your manuscript has been judged scientifically suitable for publication and will be formally accepted for publication once it meets all outstanding technical requirements.

Kind regards,

Manuel Alberto de Almeida e Sousa Falcão, M.D., Ph.D.

Academic Editor

PLOS ONE
---

## [Editor Report · Acceptance letter]

1 Jun 2021

PONE-D-20-07419R3 

Efficacy of anti-vascular endothelial growth factor agents for treating neovascular age-related macular degeneration in vitrectomized eyes 

Dear Dr. Woo:

I'm pleased to inform you that your manuscript has been deemed suitable for publication in PLOS ONE. Congratulations! Your manuscript is now with our production department. 

Kind regards, 

on behalf of

Dr. Manuel Alberto de Almeida e Sousa Falcão 

Academic Editor

PLOS ONE